# Remission of Proteinuria in a Patient Affected by Crescentic IgA Nephropathy with Rapidly Progressive Glomerulonephritis Treated by Sodium-Glucose Cotransporter-2 Inhibitors: Casual or Causal Relationship?

José C. De La Flor Merino [1,*], Jacqueline Apaza Chávez [2], Francisco Valga Amado [3], Francisco Díaz Crespo [4], Pablo Justo Avila [5], Alexander Marschall [6], Michael Cieza Terrones [7], Patricia Núñez Ramos [4] and Elisa Ruiz Cicero [1]

1 Department of Nephrology, Hospital Central Defense Gomez Ulla, Glorieta del Ejercito 1 Street, 28047 Madrid, Spain
2 Department of Nephrology, Hospital Fuenlabrada, 28942 Madrid, Spain
3 Department of Nephrology, Hospital Universitario Doctor Negrin de Gran Canarias, 35016 Las Palmas de Gran Canarias, Spain
4 Department of Anatomic Pathology, Hospital Gregorio Marañón, 28009 Madrid, Spain
5 Department of Nephrology, Moncloa University Hospital, 28008 Madrid, Spain
6 Department of Cardiology, Hospital Central Defense Gomez Ulla, 28047 Madrid, Spain
7 Teaching Coordination Unit, Faculty of Medicine, Universidad Peruana Cayetano Heredia, Lima 15012, Peru
* Correspondence: jflomer@mde.es

**Abstract:** Crescentic IgA nephropathy (IgAN) with rapidly progressive glomerulonephritis (RPGN) is often associated with rapidly declining kidney function. Up to this date, specific therapy for crescentic IgAN is still unknown. Accumulating evidence suggests that sodium-glucose co-transporter-2 inhibitors (SGLT-2i) may have a role in standard therapy of glomerular diseases. However, it is unclear at what point in the natural history of specific glomerular diseases SGLT-2i can be beneficial. We report the clinical and histological features of a patient with crescentic IgAN that presented as an RPGN, who received intensive immunosuppression and renal replacement therapeutic (RRT). At the third month, the patient presented with significant improvement in his kidney function. At that point, we decided to start dapagliflozin in addition to his renin-angiotensin system (RAS) blocker, basing our decision on its proven renal benefits such as slowing the rate of decline in kidney function and reducing albuminuria. At the eighth month, the patient's renal function gradually improved from serum Cr of 6.07 to 2.1 mg/dL; and urine albumin to creatinine ratio (UACR) declined from 5655 mg/g to 200 mg/g. The use of SGLT-2i in primary and secondary nondiabetic glomerular disease appears promising. It is crucial and necessary to accumulate more evidence for a more complete understanding of the mechanisms of the actions of SGLT-2i in non-diabetic glomerular disease.

**Keywords:** crescentic IgA nephropathy; rapidly progressive glomerulonephritis; sodium-glucose co-transporter-2 inhibitors

## 1. Introduction

Immunoglobulin A nephropathy (IgAN) is the most common primary glomerulonephritis disorder in the world [1–3]. The spectrum of clinical presentation includes the asymptomatic carrier, hematuria and/or proteinuria, recurrent macroscopic hematuria; and less frequent nephritic, nephrotic syndrome or rapidly progressive glomerulonephritis (RPGN). This last manifestation, that involves less than 10% of patients, usually has an aggressive course with histopathological findings of crescentic or extracapillary proliferative glomerulonephritis [4]. The presence of crescents indicates active inflammation and

predicts a poor prognosis in IgAN [4]. Hence, renal biopsy is key from a diagnostic and prognostic point of view, helping to individualize patients' treatment.

With regards to IgAN management, the renal-angiotensin-aldosterone system blockade with angiotensin-converting enzyme inhibitors (ACEIs) or angiotensin II receptor blockers (ARB-2) continues to play a central role. Sodium-glucose co-transporter-2 inhibitors (SGLT-2i) have recently been included in the management of non-diabetic glomerular diseases; however, there is still no experience with regards to the use of or the timing of initiation of SGLT-2 in RPGN [5]. In this case report, we describe the case of a patient with crescentic IgAN treated with pulse methyl prednisolone, oral steroids, intravenous cyclophosphamide and SGLT-2i with good clinical and biochemical response.

## 2. Case Report

We present the case of a 67-year-old caucasian male with a past medical history of arterial hypertension (HTN), well-controlled type-2 diabetes (T2DM) and a non-followed up microscopic hematuria. His medication included ARB-2, metformin, torasemide and statin. He was referred to our hospital because of rapid renal function deterioration. Three months prior to admission, his serum creatinine (Cr) levels were 1 mg/dL, estimated glomerular filtration rate (eGFR) was 77.5 mL/min/1.73 m$^2$ by Chronic Kidney Disease Epidemiology (CKD-EPI) formula, and urine albumin to creatinine ratio (UACR) was 90 mg/g. He presented with a three days history of headache, reduced urine output, lower limb oedema, uncontrolled HTN and gross hematuria. Six days earlier, he had had an episode of pharyngotonsillitis and was treated with empiric amoxicillin and ibuprofen. On admission, his blood pressure was 180/100 mmHg, pulse rate was 85/min, while body temperature was normal. He had moderate bilateral pretibial oedema and the rest of his physical examination was unremarkable. His blood tests showed a Cr level of 6.07 mg/dL and hypoalbuminemia without hypercholesterolemia. Urinalysis showed 3+ proteinuria by dipstick, and 50–100 red blood cells per high power field (RBC/HPF) with dysmorphic cells. The 24-h urine protein excretion was 6.5 g and spot urine albumin to creatinine ratio (UACR) of 5655 mg/g. Hemoglobin 10.6 g/dL, hematocrit: 31.8 %, white blood cell count: 7800/μL and platelet count: 188,000/μL. Serum electrolyte levels were within normal limits, except for hyperphosphatemia. Serum proteinogram showed increased immunoglobulins A (IgA) of 517 mg/dL (normal value: 70–400 mg/dL); with normal levels for IgG and IgM. Serum and urine protein immunofixation/electrophoresis (IFE) did not reveal any monoclonal bands. The rest of his analyses were negative including: cryoglobulins, hepatitis B and C, human immunodeficiency virus (HIV) anti-nuclear antibody (ANA), anti-double stranded DNA (dsDNA), anti-neutrophil cytoplasmic antibody (ANCA), rheumatoid factor and anti-glomerular basement membrane (GBM). Furthermore, serum complement-3 (C3) and complement-4 (C4) were normal. Other laboratory test results are shown in Table 1. Chest radiography showed cardiomegaly and lung congestion. On ultrasonography, both kidneys were of normal size, but presented increased parenchymal echogenicity.

**Table 1.** Laboratory findings on admission.

| | | Reference Range/Unit |
|---|---|---|
| WBC | 7800 | U/L |
| Hemoglobin (Hb) | 10.6 | 12–16 g/dL |
| Platelet count (PLt) | 188 | $10^3/\mu L$ |
| Reticulocytes count | 2.77 | 2–4% |
| Erythrocyte count | 3.43 | $4.2–5.8 \times 10^6/\mu L$ |
| Lactate dehydrogenase (LDH) | 258 | 135–214 IU/L |
| Coombs Test | Negative | NA |
| Total Bilirubin | 0.42 | 0.1–1 mg/dL |

**Table 1.** *Cont.*

|  |  | Reference Range/Unit |
|---|---|---|
| Total protein | 6.1 | 6.4–8.7 g/dL |
| Serum Albumin (Alb) | 2.52 | 3–5.5 g/dL |
| GOT | 21 | 5–32 IU/L |
| GPT | 12 | 5–33 IU/L |
| Triglycerides | 161 | 30–150 mg/dL |
| Total cholesterol | 192 | 110–200 mg/dL |
| Urea | 185 | 17–60 mg/dL |
| Creatinine | 6.07 | 0.6–1.2 mg/dL |
| Na | 138 | 135–145 mmol/L |
| K | 4.2 | 3.5–5.5 mmol/L |
| Cl | 99 | 95–110 mmol/L |
| P | 7.5 | 2.5–4.5 mg/dL |
| Albumin-corrected calcium | 8.78 | 8–10.4 mg/dL |
| CRP | 1.02 | 0.1–0.5 mg/dL |
| Hbs-Ag | Negative | NA |
| HCV-Ab | Negative | NA |
| HIV | Negative | NA |
| Cryoglobulins | Negative | NA |
| CFH | 250 | 225–760 µg/mL |
| Autoantibodies CFH | Negative | <18 AU/L |
| C3 nephritic factor (C3NF) | Negative | Ratio > 1.022 |
| C3 | 110 | 90–180 mg/dL |
| C4 | 25.1 | 10–40 mg/dL |
| RF | Negative | <15 IU/ml |
| ANA, Antids-DNA, ANCA and cryoglobulin | Negative | NA |
| Anti-GBM | Negative | <1 AI |
| Anti-PLA2R Ab (ELISA) | Negative | NA |
| Beta 2 microglobulin | 1.09 | <0–20 mg/dL |
| IgG | 885 | 800–1600 mg/dL |
| IgA | 517 | 70–400 mg/dL |
| IgM | 98 | 90–180 mg/dL |
| UPCR | 6350 | <20 mg/g |
| UACR | 5655 | <30 mg/g |
| Urine red blood cells | 50–100 | /HPF |
| 24-h urine total protein excretion | 6.5 | <0.15 g/24-h |
| SPEP M-protein concentration | No monoclonal band | NA |
| Urine immunofixation-electrophoresis: | No monoclonal band | NA mg/dL |

AI: Activity index, AU: arbitrary units, NA: not applicable, WBC: white blood cells, GOT: glutamate-oxaloacetate transaminase, GPT: glutamate pyruvate transaminase, Na: sodium serum, K: potassium serum, Cl: chloride serum, P: phosphate serum, CRP: C-reactive protein, CFH: complement factor H, C3NF: complement 3 nephritic factor, C3: complement 3, C4: complement 4, RF: rheumatoid factor, ANA: antinuclear antibody, Antids-DNA: anti-double stranded DNA antibody, ANCA: anti-neutrophil cytoplasmic autoantibody, Anti-GBM: anti-glomerular basement membrane, Anti-PLA2R Ab: anti-phospholipase A2 receptor antibody, Ig: immunoglobulin, UPCR: spot urine protein-to-creatinine ratio, UACR: spot urine albumin-to-creatinine ratio, SPEP: serum protein electrophoresis, HPF: high power field.

On admission, renal replacement therapy (RRT) with intermittent hemodialysis (HD) was initiated due to uremia, and a percutaneous renal biopsy was performed (as shown in Figure 1). Thirty-six glomeruli were examined under light microscopy (LM), 17 (45%) of

which were globally sclerosed. The rest of the glomeruli exhibited diffuse mesangial and endocapillary hypercellularity, and in 16 (84%) out of 19 glomeruli necrotizing crescents were found (15 cellular and 1 fibrous/fibrocellular). Capillary walls did not show double contoured appearance nor intravascular thrombi or vasculitis. A mild degree of interstitial scarring (30%) and tubular atrophy was also noted. Some tubules showed dilation of tubular lumens, flattening of epithelium, simplification and loss of brush border. No RBC casts were seen. Mild arteriolosclerosis (10%) was observed (Figure 1A–D). Immunofluorescence (IF) demonstrated diffuse global granular deposits with IgA (2+) and C3 (2+) in mesangium. Positivity for fibrinogen (1+) was seen in areas of extracapillary proliferation with fibrinoid necrosis. IgG, IgM and C1q were negative. Kappa and lambda chain stains did not show restrictions (Figure 1E). Electron microscopy (EM) revealed large electron-dense non-organized deposits in mesangium (Figure 1F). Therefore, a diagnosis of crescentic IgAN, with a MEST-C score of M0 E1 S1 T1 C2, and acute tubular necrosis (ATN) was established.

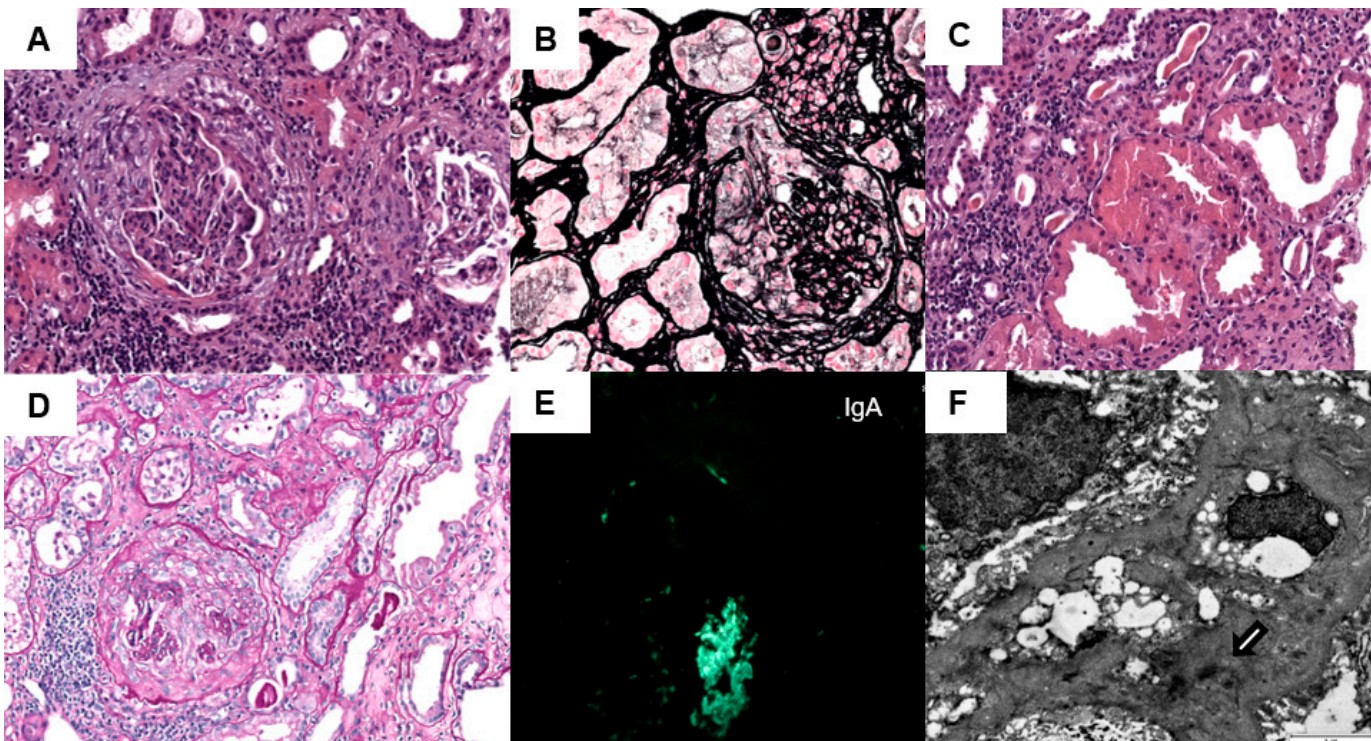

**Figure 1.** (**A**) Hematoxylin and eosin staining (×200), showing cellular crescents in two glomeruli. (**B**) Glomerulus showing a cellular crescent (Jones methenamine silver stain × 200). (**C**) tubules with dilation of tubular lumens, flattening of epithelium, simplification and loss of brush border (HE × 200). (**D**) Glomerulus showing a cellular crescent and tubules with dilated lumen and flattened regenerating epithelium; their lumens contain necrotic cellular debris (PAS × 200). (**E**) Immunoflourescence (IF) showing diffuse global granular deposits with IgA (2+) in mesangium (IF × 10). (**F**) Electronic microscopy examination showing mesangial electron-dense deposits (white arrow) (EM × 10,000).

In light of these findings, the patient was treated with intravenous methylprednisolone of 1 g for 3 days, followed by a tapering dose of oral prednisolone for the following five months. Intravenous cyclophosphamide was given at 0.5 g/m$^2$ body surface area monthly for 6 months. At the third month of follow up, his UACR decreased to 3200 mg/g and there was an improvement in his residual renal urea clearance (krU) and creatinine clearance (CrCl) in 24-h from 2.66 to 10.96 mL/min, and from 9.9 to 31.8 mL/min, respectively. Given the significant improvement in the GFR without requiring more HD sessions, we decided to add dapagliflozin (SGLT-2i) to the treatment, given its proven renal benefits such as slowing the rate of decline in kidney function and reducing proteinuria [4]. At the eighth

month follow up, the patient´s renal function gradually improved from a serum Cr of 6.07 to 2.66 mg/dL. The UACR, CrCl and krU in 24-h results were sustained to 200 mg/g, 29.6 and 7.33 mL/min, respectively (Table 2). The patient did not suffer from any urinary tract infection or other known side effects associated with taking SGLT-2i.

**Table 2.** Evolution of laboratory characteristics and renal function parameters.

| | Baseline | At 1 Month | At 2 Month | At 3 Month | At 4 Month | At 5 Month | At 8 Month |
|---|---|---|---|---|---|---|---|
| Hb (mg/dL) | 10.6 | 11.3 | 13 | 12.4 | 13 | 12 | 12.3 |
| Ferritin (ng/mL) | 233 | 224 | 599 | 468 | 556 | 372 | 437 |
| Tsat (%) | 30 | 21 | 32 | 38 | 35 | 33 | 41 |
| Serum creatinine (mg/dL) | 6.07 | 4.14 | 5.14 | 2.35 | 2.66 | 2.69 | 2.66 |
| eGFR (CKD-EPI) (mL/min) | 8.8 | 13.9 | 10.7 | 27.6 | 23.8 | 23.4 | 23.8 |
| Serum urea (mg/dL) | 185 | 123 | 103 | 76 | 99 | 89 | 88 |
| Kru (mL/min) | 2.66 | 3.44 | - | 10.96 | 6.9 | 7.4 | 7.33 |
| Urine volume (mL/24 h) | 800 | 1900 | 1600 | 1900 | 2400 | 1900 | 1600 |
| CrCl (mL/min) | 9.9 | 13 | - | 31.8 | 29.6 | 28.7 | 29.6 |
| UACR (mg/g) | 5655 | 6769 | 5599 | 3200 | 1271 | 443 | 200 |
| Urine red blood cells/HPF | 50–100 | 50–100 | - | 25–35 | - | 15–25 | 10–15 |
| Urine glucose (mg/dL) | Negative | 5 | Negative | 100 | 300 | 288 | 310 |
| Albumin-corrected calcium (mg/dL) | 8.78 | 9.1 | 9.1 | 8.5 | 9.3 | 8.9 | 8.8 |
| Phos (mg/dL) | 7.5 | 5.6 | 4.7 | 4.5 | 4.6 | 3.3 | 2.9 |
| Serum Albumin (mg/dL) | 2.52 | 2.84 | 2.97 | 3.34 | 3.11 | 3.43 | 3.69 |
| PTH (pg/mL) | 154 | 202 | 232 | - | 101 | - | 79 |
| Calcitriol (ng/mL) | 6.31 | 33.5 | 46.9 | 21 | 25 | 20.6 | 27.8 |
| Na (mmol/L) | 138 | 137 | 141 | 138 | 140 | 138 | 139 |
| K (mmol/L) | 4.2 | 4.5 | 4.2 | 4.2 | 3.9 | 4.2 | 3.7 |
| Blood pressure (mmHg) | 180/100 | 151/87 | 137/82 | 128/79 | 121/74 | 133/81 | 125/78 |
| Cyclophosphamide | 0.5 g/m$^2$ monthly | 0.5 g/m$^2$ monthly | 0.5 g/m$^2$ monthly | 0.5 g/m$^2$ monthly | 0.5 g/m$^2$ monthly | 0.5 g/m$^2$ monthly | - |
| Corticosteroids | MPD 1 g for 3 days | PDN 60 mg/day | PDN 60 mg/day | PDN 50 mg/day | PDN 30 mg/day | PDN 10 mg/day | - |
| ARB-2 | Yes | Yes | Yes | Yes | Yes | Yes | Yes |
| Dapagliflozin (SGLT-2i) | - | - | - | 10 mg/day | 10 mg/day | 10 mg/day | 10 mg/day |
| Weekly HD regimens | Thrice | once | - | - | - | - | - |

Hb: hemoglobin; Tsat: transferrin saturation; eGFR: estimated glomerular filtration rate; CKD-EPI: Chronic Kidney Disease Epidemiology formula; HPF: high power field; Kru: residual kidney urea clearance; CrCl: creatinine clearance in 24h; UACR: spot urine albumin-to-creatinine ratio; Phos: serum phosphorus; PTH: parathyroid hormone; Calcitriol; Na: sodium serum, K: potassium serum; MPD: methylprednisolone; PDN: prednisone; SGLT-2i: sodium-glucose co-transporter-2 inhibitors; HD: hemodialysis.

## 3. Discussion

Crescentic IgA nephropathy with RPGN is often associated with rapidly progressive kidney failure. This syndrome can occur at presentation or during the course of IgAN with or without clinical features of IgA vasculitis [6]. RPGN is defined as a ≥50% decline in eGFR over three months or less, where reversible causes have been excluded according to the 2021 Kidney Disease: Improving Global Outcomes (KDIGO) glomerulonephritis guidelines [7]. In our case, the clinical presentation was highly suggestive of IgAN, but the presence of severe acute kidney injury (AKI) prompted us to rule out other causes of RPGN, such as ANCA-associated vasculitis, anti-GBM disease, lupus nephritis, extracapillary glomerulonephritis, C3 glomerulopathy, or infection-associated glomerulonephritis, tubulointerstitial nephritis (TIN) immune-mediated and ATN from red blood cell cast obstruction and/or heme toxicity. The kidney biopsy confirmed our suspected diagnosis of crescentic IgAN with ATN.

Histologically, IgAN is described according to the Oxford scale (MEST-C), being the most used and validated. Each of these variables correlates independently with the renal prognosis, especially the presence of glomerular sclerosis (S) and tubular fibrosis (T) [8]. Crescentic IgAN was defined as IgAN with the presence of crescents in more than 50% of glomeruli observed in the kidney biopsy. Likewise, the presence of crescents is directly proportional to the extent of the AKI [6]. These histological findings, in addition to clinical and demographic variables, predict a risk of deterioration in renal function of up to 50%,

known as "International IgAN Prediction Tool" [9], that currently is the recommended method of risk stratification for IgAN in the 2021 KDIGO glomerulonephritis guidelines [7]. However, there is a lack of serum or urinary prognostic biomarkers to stratify and identify patients with a higher risk of renal deterioration beyond the proteinuria and renal function for follow-up. Our patient had a risk of 50% decline in eGFR or progression to end stage kidney disease (ESKD) at 5 years after renal biopsy of 66.76%, without including the presence of crescents. As previously mentioned, their presence on its own is a distinctive pathologic feature associated with disease severity [10]. Our patient showed a C2 score in the kidney biopsy. The extensive crescent formation demonstrated poor kidney outcomes despite the use of immunosuppression, as more than two-thirds of patients with this degree of severity develop ESKD [11]. Based on these findings, Haas et al. [4], in a large IgA nephropathy cohort pooled from four retrospective studies, addressed crescents as a predictor of renal outcomes and determined whether the fraction of crescent-containing glomeruli associates with survival from either a 50% decline in eGFR or ESRD (combined event) adjusting for covariates used in the original Oxford study. With more than 3000 subjects studied, 36% of them had cellular or fibrocellular crescents. According to their results, having crescents in at least one quarter of the glomeruli was independently associated with a combined event in patients receiving and not receiving immunosuppression. For this reason, they proposed adding a crescent score to the Oxford classification: C0 (no crescents); C1 (crescents in <25% glomeruli), identifying patients at increased risk of poor outcome without immunosuppression; and C2 (crescents in >25% glomeruli), identifying patients at even greater risk of progression, even with immunosuppression.

To date, a specific therapy for crescentic IgAN is not available. The KDIGO 2021 glomerulonephritis guidelines for the management of IgAN recommend that either the presence or the relative number of crescents should neither be used to determine the progression of IgAN nor the choice of immunosuppression [7]. In addition, KDIGO also proposed that the presence of crescents in >50% of glomeruli in a kidney biopsy with no decline in GFR does not constitute a rapidly progressive situation, and thus, immunosuppressive therapy is not indicated [12]. In contrast, KDIGO 2021 glomerulonephritis guidelines recommend aggressive immunosuppressive therapy with cyclophosphamide and corticosteroids in patients with crescentic IgAN and RPGN, in accordance with the guidelines for ANCA-associated vasculitis [7].

SGLT-2i acts in the proximal tubule to inhibit sodium and glucose reabsorption via SGLT-2, and has renoprotective effects independently of its glucose-lowering effects [13–16], including proteinuria reduction, likely because of an intraglomerular perfusion pressure reduction [17]. However, beneficial effects on kidney outcomes in clinical trials are independent of reduction in proteinuria [18]. In the Dapagliflozin and Prevention of Adverse Outcomes in Chronic Kidney Disease (DAPA-CKD) trial, dapagliflozin reduced the risk of kidney failure and prolonged survival in participants with chronic kidney disease with and without type 2 diabetes (T2DM), including those with IgAN [13]. The post-hoc analysis of the DAPA-CKD study demonstrated that in patients with IgAN, when added to ACEi/ARB-2 therapy, dapagliflozin significantly and substantially reduces the risk of CKD progression with a favorable safety profile [19]. Interestingly, patients who were receiving active immunosuppression or suffered from lupus nephritis and ANCA-associated vasculitis were excluded. The results of this study suggest that SGLT-2i may play a role in standard therapy of glomerular diseases [19]. However, the timing of initiation of SGLT-2i in specific glomerular diseases to maximize therapeutic benefits remains unclear [20]. Emily et al. [18], recommend the addition of SGLT-2i to renin-angiotensin system (RAS) blockers, in patients with glomerular disease with proteinuria in whom prompt remission is less likely.

In our case, all the measures of the first and second therapeutic steps were applied, associated with the previously mentioned immunosuppression. We believe that in those patients with an immune-mediated primary glomerulonephritis, where the aberrant immune response is the central pathologic mechanism, as is the rapidly progressive forms of IgAN, management should focus on prompt and aggressive immunosuppression with the

aim of inducing a rapid remission [21]. Moreover, when improvement of renal function is sustained, we suggest that SGLT-2i should be added to standard therapy, so that the patients can benefit from its antiproteinuric and cardio-nephroprotective effects. However, despite the fact that their use is becoming more widespread, and although SGLT-2i are safe and are likely to play an important role in the management of IgAN or other primary and secondary non-diabetic glomerular diseases in the future, their benefits beyond standard interventions are unclear. To the best of our knowledge, to date, there are no case reports or series describing the use of SGLT-2i in patients with crescentic IgAN. Although the evidence generated by the case report or series has a low internal validity, our case description would be interesting for accumulating evidence and providing a clearer scenario on its use. Finally, cumulative evidence suggests that SGLT-2i may act on many different types of kidney cells (endothelial, mesangial cells and podocytes) and in distinct kidney compartments (tubulointerstitial fibrosis) with specific molecular and cellular effects [22].

## 4. Conclusions

The use of SGLT-2i in primary and secondary glomerular diseases (whether this is immune or nonimmune mediated) unrelated to diabetes, appears to be a promising therapy for optimized supportive care. There is still a long way to go to fully exploit the potential of these drugs; it is crucial and necessary to accumulate more evidence for a more complete understanding of the mechanisms of action of SGLT-2i in non-diabetic glomerular disease.

**Author Contributions:** Conceptualization, J.C.D.L.F.M. and J.A.C.; methodology, J.C.D.L.F.M., J.A.C., F.V.A., F.D.C., P.J.A.; software, A.M., P.N.R., M.C.T.; writing—original preparation, J.C.D.L.F.M., J.A.C., M.C.T., F.V.A., P.N.R. and P.J.A.; writing—review and editing, J.C.D.L.F.M., A.M., P.J.A., M.C.T., F.V.A. and E.R.C.; visualization, F.D.C. and P.N.R.; supervision, E.R.C. and M.C.T. All authors have read and agreed to the published version of the manuscript.

**Funding:** This research received no external funding.

**Institutional Review Board Statement:** Ethical review and approval were waived for this study as per our Institutional Review Board, due to the nature of our research as a case report.

**Informed Consent Statement:** Written informed consent has been obtained from the patient to publish this paper (including the publication of images).

**Data Availability Statement:** No new data were created or analyzed in this study. The data used to support the findings of this study are available from the corresponding author on request (Contact J.D.F., josedelaflor81@yahoo.com, jflomer@mde.es).

**Conflicts of Interest:** The authors declare no conflict of interest.

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
