# Peer review of "Remission of Proteinuria in a Patient Affected by Crescentic IgA Nephropathy with Rapidly Progressive Glomerulonephritis Treated by Sodium-Glucose Cotransporter-2 Inhibitors: Casual or Causal Relationship?"

_kidneydial, doi:10.3390/kidneydial2040049_

Round 1
Reviewer 1 Report
I have no observations in relation to the case description and reporting of relevant data either at baseline or after 6 months follow up, but:
1) The first part of references is about prediction of outcomes in crescentic IgA nephropathy, but nor in discussion neither in conclusions i could find an expicit referral to this item. So i suggest to include this item in discussion.
2) The AA conclude that: 'The use of SGLT-2i in primary and secondary glomerular diseases (whether this is immune or nonimmune mediated) unrelated to diabetes appears to be a promising therapy for optimized supportive care'.
Neither in introduction nor in discussion i could find the citation of other case report or case series that could have been helpful to support these conclusions.
Even if the evidence generated by case report or series has a low internal validity, it would be intereseting to cumulate such evidence and individual patient data in order to offer a clearer scenario.
So i suggest to the AA to include this item in discussion and conclusions
Author Response
Reviewer 1 report:
Comments to authors
I have no observations in relation to the case description and reporting of relevant data either at baseline or after 6 months follow up, but:
1) The first part of references is about prediction of outcomes in crescentic IgA nephropathy, but nor in discussion neither in conclusions i could find an expicit referral to this item. So i suggest to include this item in discussion.
2) The AA conclude that: 'The use of SGLT-2i in primary and secondary glomerular diseases (whether this is immune or nonimmune mediated) unrelated to diabetes appears to be a promising therapy for optimized supportive care'.
Neither in introduction nor in discussion i could find the citation of other case report or case series that could have been helpful to support these conclusions.
Even if the evidence generated by case report or series has a low internal validity, it would be intereseting to cumulate such evidence and individual patient data in order to offer a clearer scenario.
So i suggest to the AA to include this item in discussion and conclusions
Author Response:
Thank you very much for your recommendation.
Point 1:
Based on the reviewer's first comment, we have modified the last sentence of the second paragraph of the discussion to include the point about prediction of outcomes in crescentic IgA nephropathy
We have deleted this sentence ¨Based on these findings, Haas et al.[4], proposed adding the crescent score to the Oxford classification¨ and changed it to another sentence:
Based on these findings, Haas et al. [4], in a large IgA nephropathy cohort pooled from four retrospective studies, they addressed crescents as a predictor of renal outcomes and determined whether the fraction of crescent-containing glomeruli associates with survival from either a 50% decline in eGFR or ESRD (combined event) adjusting for covariates used in the original Oxford study. With more than 3000 subjects studied, 36% of them had cellular or fibrocellular crescents. According to their results, having crescents in at least one fourth of the glomeruli was independently associated with a combined event in patients receiving and not receiving immunosuppression. For this reason, they proposed adding crescent score to the Oxford classification: C0 (no crescents); C1 (crescents in < 25% glomeruli), identifying patients at increased risk of poor outcome without immunosuppression; and C2 (crescents in > 25% glomeruli), identifying patients at even greater risk of progression, even with immunosuppression.
In addition, we have deleted this sentence ¨C2 score is defined as the presence of cellular/fibrocellular crescents in ≥ 25% of glomeruli¨, in order to not repeat the classification discussed in the following sentence.
Point 2:
Regarding point 2, we agree fully with the reviewer. There is currently no literature such as series or case reports that allow us to draw these conclusions, only the results obtained in the post-hoc analysis of the DAPA-CKD study that demonstrated the benefit of the use of SGLT-2i in patients with IgAN. For this reason, we consider it appropriate to add a comment at the end of the last paragraph of the discussion and conclusions on the absence of evidence in this item and thus improve the discussion of our case:
¨ However, despite the fact that their use is becoming more widespread, and although SGLT-2i are safe and are likely to play an important role in the management of IgAN or other primary and secondary non-diabetic glomerular diseases in the future, their benefits beyond standard interventions are unclear. To the best of our knowledge, to date, there are no case reports or series describing the use of SGLT-2i in patients with crescentic IgAN. Although the evidence generated by case report or series has a low internal validity, our case description would be interesting to accumulate evidence and provide a clearer scenario on its use. ¨
We have removed this sentence ¨Hence, more studies are needed to assess the efficacy and safety of SLGT-2i in nondiabetic glomerular disease¨ from the conclusions and changed it to another, according to the reviewer's recommendations:
¨There is still a long way to go to fully exploit the potential of these drugs; it is crucial and necessary to accumulate more evidence for a more complete understanding of the mechanisms of action of SGLT-2i in non-diabetic glomerular disease¨.

Reviewer 2 Report
In this paper authors present a patient with crescentic IgA nephropathy and rapidly progressive glomerulonephritis who was treated with renin-angiotensin blockers and dapagliflozin, an SGLT-2 inhibitor. Patient showed improved renal function after five months of this treatment.
This case report is interesting, it shows the necessary data. Use of English language is appropriate, the paper is easy to understand. I would suggest only mild English language editing, there are some misspellings.
Author Response
Reviewer 2 report:
Comments to authors
In this paper authors present a patient with crescentic IgA nephropathy and rapidly progressive glomerulonephritis who was treated with renin-angiotensin blockers and dapagliflozin, an SGLT-2 inhibitor. Patient showed improved renal function after five months of this treatment.
This case report is interesting, it shows the necessary data. Use of English language is appropriate, the paper is easy to understand. I would suggest only mild English language editing, there are some misspellings.
Author Response:
Thank you very much for your recommendation, we have thoroughly revised the English language of our manuscript with the help of a colleague who is proficient in English writing.
If in any case further revision is needed, we will use the English editing service of the journal. Thank you very much.

Round 2
Reviewer 1 Report
Satisified by the reply